# Grapevine Cane Extracts: Raw Plant Material, Extraction Methods, Quantification, and Applications

**DOI:** 10.3390/biom10081195

**Published:** 2020-08-17

**Authors:** María José Aliaño-González, Tristan Richard, Emma Cantos-Villar

**Affiliations:** 1Instituto de Investigación y Formación Agraria y Pesquera (IFAPA), Consejería de Agricultura, Ganadería, Pesca y Desarrollo Sostenible, Rancho de la Merced, Ctra. Cañada de la Loba, CA-3102 km 3.1, 11471 Jerez de la Frontera, Spain; mariaj.aliano@juntadeandalucia.es; 2Université de Bordeaux, ISVV, EA 3675 Groupe d’Etude des Substances Végétales à Activité Biologique, 33882 Villenave d’Ornon, France; tristan.richard@u-bordeaux.fr

**Keywords:** grapevine cane extract, stilbenes, grapevine, extraction, antioxidant, preservative, biostimulant, antifungal, bioplaguicide

## Abstract

Grapevine canes are viticulture waste that is usually discarded without any further use. However, recent studies have shown that they contain significant concentrations of health-promoting compounds, such as stilbenes, secondary metabolites of plants produced as a response to biotic and abiotic stress from fungal disease or dryness. Stilbenes have been associated with antioxidant, anti-inflammatory, and anti-microbial properties and they have been tested as potential treatments of cardiovascular and neurological diseases, and even cancer, with promising results. Stilbenes have been described in the different genus of the *Vitaceae* family, the *Vitis* genera being one of the most widely studied due to its important applications and economic impact around the world. This review presents an in-depth study of the composition and concentration of stilbenes in grapevine canes. The results show that the concentration of stilbenes in grapevine canes is highly influenced by the *Vitis* genus and cultivar aspects (growing conditions, ultraviolet radiation, fungal attack, etc.). Different methods for extracting stilbenes from grapevine canes have been reviewed, and the extraction conditions have also been studied, underlining the advantages and disadvantages of each technique. After the stilbenes were extracted, they were analyzed to determine the stilbene composition and concentration. Analytical techniques have been employed with this aim, in most cases using liquid chromatography, coupled with others such as mass spectrometry and/or nuclear magnetic resonance to achieve the individual quantification. Finally, stilbene extracts may be applied in multiple fields based on their properties. The five most relevant are preservative, antifungal, insecticide, and biostimulant applications. The current state-of-the-art of the above applications and their prospects are discussed.

## 1. Introduction

Wine production is an important part of agriculture and the beverage industry worldwide. In fact, according to the latest officially-recorded data from the Organisation Internationale de la Vigne et du Vin (OIV), the global consumption of wine in 2016 was 24,144,400,000 L [1].

Winemaking is a multistage process producing a huge amount of organic and inorganic waste, which is usually discarded, or used as compost or animal food. New uses have been proposed in recent years, including its transformation into chemicals, bioproducts, dyes, etc. [2,3,4,5]. However, its biochemical conversion is an time-consuming and costly process and a large area of the vineyard is required to generate a worthwhile amount of fuel, chemicals, etc. [6,7]—the reason why these alternatives have not already been incorporated in most vineyards.

It is possible to distinguish two main categories of winery waste: that generated during the collection and that resulting from the winemaking process. During winemaking—in the first stage for white wines and after alcoholic fermentation for reds—must/wine is crushed in a pneumatic press, producing a solid residue known as *pomace*. The amount of pomace generated depends on the grape variety, the cultivation conditions, and the pressing conditions used, but many researchers have concluded that pomace represents around 20–30% of grape weight [8,9]. Lees are another kind of winemaking waste. They consist of yeast biomass, undissolved carbohydrates of cellulosic nature, lignin, proteins, phenolic compounds, tartrates acid salts and fruit skins, grains, and seeds in suspension [10] produced in the tanks during the alcoholic fermentation process. Wine lees are usually at a concentration of 5% *v*/*v* [6] and they are distilled to recover ethanol or elaborate distilled beverages [11].

In the collection process, the major by-products of vineyards are *grapevine canes* (also called stems, shoots, or stalks), with an average production of around 2–5 tons per hectare and year [12]. Grapevine canes are rich in lignin, cellulose, nitrogen, and potassium—the reason why they are highly composted in the field or burned [13]. However, they also present high contents of interesting compounds such as polyphenols, proteins, and stilbenes [14]. Stilbenes are an interesting family of non-flavonoid polyphenols that belong to the phenylpropanoid group. They are produced by plants in response to biotic and abiotic stresses [15]. The *trans (t)* or *cis* (*c*) ethene double bond and the different radicals that can bond to phenyl structures mean that stilbenes constitute a huge and varied group of molecules [15,16,17]. This family is composed of monomers and oligomers. Monomers are modified by the different radicals and steric configurations of the molecules, whereas oligomers correspond to varied condensation from the resveratrol monomer (dimer, trimer, etc.) or to monomer hydroxylation, methylation, and glycosylation processes.

Several researchers have reported the important health-related properties of these compounds. Stilbenes have shown positive results in the treatment of cancer, decreasing cancer cell proliferation in some cases [18,19,20,21]. Some stilbene compounds have exhibited a decisive role as chemoprevention agents, inhibiting tumor initiation, tumor promotion, and the progression of malignant cells in breast [22], bladder [23], colon [24], and gastric [25] cancers, and even leukemia [26].

Furthermore, stilbenes present significant anti-inflammatory activity in the brain, which represents crucial progress in the treatment of neurodegenerative diseases such as Alzheimer’s [27,28,29]. Stilbenes’ potential anti-inflammatory activity is based on the inhibition of enzymes that activate cytokines [30]. These results have important cardioprotective applications, which have been suggested in the so-called “French paradox,” which explains the low incidence of coronary heart disease among French people consuming a diet rich in saturated fats but with a high consumption of wine (a source of stilbenes) [31,32].

Finally, studies of resveratrol (the most relevant stilbene) have reported increases in the maximum lifespans of *Saccharomyces cerevisiae*, *Caenorhabditis elegans* [33,34,35,36], the fruit fly *Drosophila melanogaster*, and the honey bee *Apis mellifera* [35,37,38]. Regarding mammals, a dietary supplement with resveratrol was supplied to obese mice, the results showing that their lifespan was longer and healthier compared with the control mice [39].

Thus, stilbenes show important health benefits that could be applied in a variety of fields, such as agriculture, cosmetics, nutraceuticals, and medicine [40,41,42,43,44,45]. Grapevine canes are a rich stilbene source that is usually discarded. For this reason, recent studies have investigated the process for extracting stilbenes from grapevine canes, and the influences of the *Vitis* species and climate conditions on stilbene content. Besides, multiple analytical techniques have identified and quantified stilbenes from these grapevine cane extracts. This research not only aimed to gain more in-depth knowledge of these compounds and their presence in vineyard by-products, but has also proposed numerous applications for extracts that could contribute to sustainability in viticulture since grapevine canes are currently a waste product without any added-value.

## 2. Raw Plant Material: Grapevine Cane

The 1000 stilbenes described in the plant kingdom [46,47,48,49] have been isolated from diverse plant families, such as pine (*Pinaceae*), cypress (*Cyperaceae*), peanut (*Fabaceae*), sorghum (*Poaceae*), and grape (*Vitaceae*) [16,50,51].

*The vitaceae* family is formed by 900 species with 17 genera [52,53], and stilbenes have mainly been described in five of those genera: *Ampelopsis, Cissus, Cyphostemma, Parthenocissus,* and *Vitis*. The *Vitis* genus is one of the most studied genera due to its economic impact. Research has identified stilbenes in different *Vitis* species and diverse plant parts. A summary of the identified and quantified stilbenes from grapevine canes can be found in Table 1, which includes the stilbene concentrations found. 

The composition and concentration of stilbenes are highly related to the *Vitis* genus and cultivar aspects (growing conditions, ultraviolet irradiation, mechanical injury, chemical presence, etc.). For example, in the same climate conditions, the most resistant genera to diseases show the highest concentrations of stilbenes. since they are involved in the resistance responses of plants to pathogen infections [63,71]. Moreover, growing and climate adversities usually result in an increase in the stilbene concentration because they are phytoalexins, and are thus produced as a response to stressful conditions [72,73].

*Vitis vinifera* is one of the better-known species because it has been used in a variety of applications for years. Many stilbenes have been characterized in it, and a huge range of concentrations has been described. This variability may be due to a wide range of factors, including variety, cultivation zone, grapevine management, and extraction method [74,75,76]. Moreover, different stilbenes were found at different concentrations in winter-buds, shoot internodes, surface roots of the rootstock, shoot tips with young leaves, and a tendril at the beginning of the leaf expansion, inflorescence at full bloom, cluster at veraison, cluster stems, grapevine canes, and berry skins and seeds from Vitis Vinifera [14]. The chemical structures of these stilbenes can be found in Figure 1.

Altogether, comparing the stilbene concentration according to the different *Vitis* species is complicated as the grapevine canes were not extracted and analyzed under the same conditions in all cases. Moreover, some stilbenes have not been quantified and the same stilbenes have not been studied in all the species. However, the following conclusions can be drawn (Table 1): *V. amurensis* presented a high concentration of total stilbenes, mainly *t*-piceatannol, *t*-resveratrol, and ampelopsin E. *V. rupestris* also showed high stilbene content, with a high ω-viniferin concentration. *V. riparia* stood out for its high ε-viniferin content. In fact, *Vitis amurensis, Vitis riparia,* and *Vitis ruperstris* have exhibited the highest stilbene concentrations, related to the fact that they are the most resistant genera to fungal diseases [63].

Regarding grapevine canes, numerous varieties of *Vitis vinifera* have been examined and many extraction methodologies have been tested. For this reason, a range of concentrations have been included in Table 1, but specific data about stilbene content according to the extraction methodology and variety can be seen in Table 2. Despite the variability found, some varieties may be suggested as the highest stilbene producers, such as Pinot noir and Gewurztraminer, as suggested by Guerrero et al. [70], in agreement with other authors [14,69,76,77,78] (Figure 2).

The concentrations found are directly related to vine factors such as variety, climate, and growing conditions—or even the presence of a pathogen. In addition, the possibility of using external factors such as time of storage has been suggested to increase the stilbene content in grapevine cane samples. Gorena et al. [76] studied the influence of post-pruning storage on stilbene content. To this end, pruned grapevine canes were stored at room temperature for eight months and their stilbene contents were measured. A significant increase in stilbene concentration, especially in the *t*-resveratrol level, was reported during the storage process. The authors concluded that results were greatly influenced by variety, but storage of approximately three months after pruning could result in five-fold increases in stilbene content.

Cebrián et al. [79] and Houillé et al. [75] supported these results, suggesting that six weeks was enough time to increase stilbene production. Besides, Houillé extended the post-pruning investigation to include the storage temperature as a variable. The results suggested that stilbene accumulation was directly related to this factor, temperatures between 15 and 20 degrees being the most favorable for stilbene increases. On the other hand, Soural et al. [80] also studied the influence of the storage time on the stilbene content. In this case, when the storage process was up to 37 weeks, the stilbene content increased, especially regarding *t*-resveratrol, *t*-ε-viniferin, and r2-viniferin. However, long periods of storage, such as a year, gradually reduced the stilbene concentration.

Billet et al. [69] proved that mechanical stress represents an influential storage factor as well. In this case, mechanical wounding on freshly-pruned canes caused a significant increase in stilbene content, the maximal stilbene accumulation being reached in just two weeks instead of the six weeks required for the control samples. Toasting in mild conditions has shown a positive effect on stilbene production in grapevine canes. Sánchez-Gómez et al. [81] crushed grapevine canes from two varieties, and the samples were subjected to four kinds of treatment before their analysis: control (stored at room temperature), light toasting, medium toasting, and high toasting. The toasting process took place in an air circulation oven at a constant temperature of 180 degrees for three different lengths of time: 15 min (light), 30 min (medium), and 45 min (high). Light toasting resulted in significant increases in stilbene content, whereas medium and high toasting decreased the concentration, the samples presenting lower concentrations than the non-toasted ones (control).

To sum up, canes from varieties such as Pinot noir and Gewurztraminer can be proposed as suitable for obtaining stilbene extract. After pruning, several weeks of storage at 20 °C followed by a light toasting process are recommended to increase the stilbene contents in canes, and therefore to achieve the richness in the extracts.

## 3. Extraction Methods

A pre-processing treatment involving either cutting the samples into 10–20 cm pieces or crushing them takes place before extraction. Soural et al. [66] conducted a comparative study of the two pre-processing methodologies with solid–liquid extraction and the results showed higher stilbene concentrations when the grapevine canes were crushed instead of cut. Next, the humidity is removed from the grapevine canes. In general, samples are dried in ovens or chambers at temperatures around 40 °C until weight stabilization, or lyophilized and stored under controlled temperature and humidity conditions before the analysis.

Regarding the solvent, the literature describes the alcohols (methanol or ethanol) from the protic group as the best solvent for extracting stilbenes from grapevine canes [77], although acetone (aprotic solvent) has shown good results too [74]. In general, it was concluded that extractions using methanol resulted in higher concentrations of *t*-resveratrol, whereas extractions with acetone presented higher concentrations of *t*-ε-viniferin and r2-viniferin [66]. Moreover, the use of alcohol distillates from the wine industry for stilbene extraction has been also studied. The methodology applied by Rodríguez-Cabo et al. [55] showed that the stilbene (*t*-resveratrol, *t*-piceid, *t*-piceatannol, and *t*-ε-viniferin) concentration extracted with alcohol distillates was similar and even higher to that recovered when a different ethanol:H_2_O ratio was used. However, this has only been tested with pressurized liquid extraction and Soxhlet methodologies on *t*-resveratrol, *t*-piceid, *t*-piceatannol, and *t*-ε-viniferin—the study not having been extended to the remaining stilbene compounds. The details of the methodologies used for extracting stilbenes from grapevine canes can be found below.

### 3.1. Solid–Liquid Extraction Methods

One of the most used methods for extracting stilbenes from grapevine canes is solid–liquid extraction due to its low cost and simplicity. A range of conditions have been used with this technique, but three methods are the most generally applied: Vergara et al. [78], Gorena et al. [76], and Lambert et al. [74].

The Vergara et al. [78] methodology (method)*,*
Table 2) uses 20 milliliters of ethanol:H_2_O (80:20 *v*/*v*) added to 2 g of sample. The extraction is performed with an ultrasonic bar at 50 Hz for five minutes. Ewald et al. [64] used this methodology for the measurement of *t*-resveratrol and *t*-ε-viniferin in different *Vitis vinifera* varieties, achieving the highest concentration in the Pinot blanc and Sauvignon blanc varieties.

The Gorena et al. [76] method (method 2, Table 2) uses 2 g of the sample extracted with 16 milliliters of ethanol:H_2_O (80:20 *v*/*v*), with an ultrasonic homogenizer at 50 Hz for 5 min. This methodology was also applied by Sáez et al. [82] to calculate the stilbene concentration in Pinot noir grapevine canes. It was detected that methods 1 and 2 only differed in the amount of solvent and the kind of ultrasonic technology employed; however, with method 1 only *t*-resveratrol and *t*-ε-viniferin were quantified, whereas with method 2 other stilbenes such as ampelopsin A, piceatannol, hopeaphenol, and r-viniferin were also detected (Table 2).

The Lambert et al. method [74] (method 3, Table 2) uses 200 mg of the sample to be extracted with 10 mL of acetone:H_2_O mixture (60:40 *v*/*v*) overnight. Subsequently, the extracts are centrifuged and the supernatant is evaporated until dryness. Finally, the dry extract is resolved in methanol/water (1:1 *v*/*v*) and filtered. Others have applied this extraction method to calculate the stilbene concentration in grapevine canes of different *Vitis vinifera* cultivars [70]. This method detected the following stilbenes: *t*-resveratrol, *t*-ε-viniferin, piceatannol, *t*-piceid, miyabenol C, r-viniferin, ampelopsin A, and *t*-ω-viniferin.

As can be observed, solid–liquid extraction has frequently been applied thanks to its many advantages: simple operation, simple apparatus, low investment cost, and high temperature and sonication pressure that facilitate the solubility and penetration of the solvent. However, it also presents disadvantages: (i) low selectivity, (ii) small enrichment factor for the analyte, (iii) a possible loss of intensity in the sample by interaction with solvent molecules, (iv) a large solvent volume, (v) a filtration step, and (vi) repeated extractions are often necessary. For these reasons, other extraction techniques have been explored.

### 3.2. High-Pressure Methods

In recent years, analytical techniques involving the high-pressure extraction of stilbenes have been applied with the aim of using low temperatures to ensure that the process is highly efficient.

Zachová et al. [67] (method 4, Table 2) proposes the use of (a) supercritical fluid extraction, and (b) pressurized liquid extraction. In the case of the former, 10 g of cut grapevine canes were placed in a supercritical fluid extraction column apparatus. The extraction column was heated to 50 °C and pressurized to 30 MPa. Ethanol was added as a polar modifier to increase the solubility of the phenolic compounds. In the case of the pressurized liquid extraction, the authors used four grams of grapevine cane placed in the extraction column with layers of glass beads before heating the column to 100 °C. Three different solvents were tested (acetone, methanol, and ethanol), and the system was pressurized to 10 MPa. Both methodologies were employed by Zachová et al. to study the stilbene concentration in the Cabernet Moravia variety. Ethanol exhibited the highest stilbene concentration when it was employed as an extraction solvent. In addition, as previously explained, Rodríguez-Cabo et al. [55] utilized pressurized liquid extraction to compare the use of alcoholic distillates and organic solvents in the stilbene extraction of a mixed grapevine cane matrix.

Supercritical fluid extraction and pressurized liquid extraction were shown to be suitable methods for the extraction of stilbenes from grapevine canes. In fact, Zachová et al. [67] conducted a comparative study to quantify three stilbenes in the Cabernet Moravia variety using supercritical fluid extraction, pressurized liquid extraction, and Soxhlet extraction. Higher stilbene concentrations were obtained by extraction with pressurized liquid and supercritical fluid compared with Soxhlet extraction. However, temperatures higher than 100 °C induced r2-viniferin degradation, which suggests that temperature is a critical factor in this kind of extraction. In contrast, the time of analysis was shorter, and the solvent consumption lower, presenting some advantages with regard to the solid–liquid extraction method. Extraction with a supercritical fluid required a high percentage of ethanol as a co-solvent to CO_2_ to collect a similar concentration of stilbenes in grapevine canes to that obtained with the solid–liquid method.

In general, high-pressure methodologies exhibit important advantages: fast and selective extraction (based on the varied solvation strength); they are automated systems; fractionation capacity; and no filtration is required. In addition, high temperatures are not required, thereby decreasing the risk of damage to thermolabile compounds. Finally, the possibility of using CO_2_ as a single solvent represents an environmental-friendly method, as CO_2_ is chemically inert, non-toxic, and non-flammable, and it reduces oxidation reaction and is easily removed from the product.

On the other hand, this technique involves high costs and energy consumption. It is for analytical use only and trained users are required due to the technical complexity. In addition, higher pressures are required to extract compounds of high molecular weight and the technique presents the risk of system clogging. Finally, CO_2_ does not elute very polar or ionic compounds, so an organic modifier solvent such as ethanol is required.

### 3.3. Microwave-Assisted Extraction Method

Microwave-assisted extraction has also been proposed as an alternative extraction technique for stilbenes in grapevine canes. In this case, Piñeiro et al. [69] (method 5, Table 2) recommended the extraction of 1 g of grapevine cane with this technique at 125 °C with a solvent ratio of 1:100 and using ethanol:H_2_O (80:20 *v*/*v*) as the solvent for five minutes. Two successive extraction steps with fresh solvent were suggested to collect the most stilbenes. A total of 29 grapevine canes from different *V. vinifera* varieties were analyzed and the average results were compared with the solid–liquid extraction method (3) [70]. The stilbene concentrations obtained were lower than those obtained using method 3, especially *t*-resveratrol.

In comparison with the previous methodologies, microwave-assisted extraction avoids air-borne contamination, and thus no hazardous fumes are produced. On the contrary, a single-step methodology excludes the possibility of solvent addition.

### 3.4. Subcritical Water Extraction Method

The organic solvents used in the main extraction methods described have negative impacts on the environment and human health [83]. Methanol has been classified as a CMR (carcinogenic, mutagenic, or toxic for reproduction product) [84]. For this reason, some researchers have proposed alternative methods such as subcritical water extraction. Gabaston et al. [85] (method 6) suggested the use of 5 g of grapevine cane powder that was extracted in a cartridge with a fiberglass filter in an accelerated solvent extraction apparatus. The extraction was carried out at 160 °C for 30 min with only one cycle, a pressure of 100 bars, a rinse volume of 30%, and a purge of 100 s. The results were compared with an extraction performed with the same technique but using ethanol:H_2_O (60:40 *v*/*v*) as the solvent, a temperature of 60 °C, and an extraction time of 5 min, yet the same cycle, pressure, rinse volume, and purge conditions. More stilbenes were found in grapevine canes using the Gabaston et al. methodology. Apart from the major compounds *t*-resveratrol and *t*-ε-viniferin, other simple compounds detected under subcritical water conditions were: piceatannol and piceid; other dimers (ampelopsin A, ampelopsin F, pallidol, parthenocissin A, and ω-viniferin); a trimer (miyabenol C); and tetramers (viniferol E, ampelopsin H, hopeaphenol, isohopeaphenol, r-viniferin, and r2-viniferin). Thus, this is an eco-friendly and automated method, in which no filtration is needed. It has the capacity to extract oligomers. However, the analysis time is long, and it requires a specific accelerated solvent extraction system, which is not available in all laboratories.

Each methodology presents different advantages and disadvantages, and in view of the fact that the same varieties were analyzed by different methodologies, it is not possible to propose a single extraction method. However, one of the most widely used methods nowadays is solid–liquid extraction since it is cheap and fast, it involves no specific training, and the materials needed are available in most laboratories. After extraction, several techniques have been used for analytical purposes to identify and characterize the stilbene compounds present in the grapevine canes, as discussed in the following section.

## 4. Grapevine Cane Extract Analysis

Once grapevine cane extracts have been obtained by the selected methodology, it is necessary to analyze them to determine the exact stilbene composition. To that end, many analytical techniques are described in detail below.

### 4.1. High-Pressure Liquid Chromatography (HPLC)

One of the most widely used methodologies for the detection and quantification of stilbenes in grapevine cane extracts is high-pressure liquid chromatography (HPLC) coupled to a photodiode array detector (DAD) and a fluorescence detector (FLD).

Soural et al. [66] (Analysis 1, Table 3) used HPLC to quantify different stilbenes in grapevine canes from *Vitis Vinifera*. To this end, they used a C18 column; as the mobile phase A the authors selected 5% acetonitrile in pure water + 0.1% *o*-phosphoric acid, and as phase B 80% acetonitrile in water + 0.1% *o*-phosphoric acid. The DAD wavelengths employed were 220 and 315 nm and the scanning range was 190–600 nm, whereas the FLD detector used an excitation wavelength of 315 nm, an emission wavelength of 395 nm, and emission scanning in the range of 300–600 nm. This methodology was applied by Zachová et al. [67] to quantify stilbenes in the Cabernet Moravia variety.

Ewald et al. [64] (Analysis 2, Table 3) used the same solid–liquid extraction methodology as Vergara et al. to obtain stilbene rich extracts, but for the identification and quantification the authors selected an HPLC-ESI-MS/MS system. For the quantification, a method with HPLC–DAD was performed. The column was C18 at 25 °C, whereas the mobile phases consisted of 1% aqueous acetic acid (*v*/*v*) (A) and methanol (B). The wavelengths selected in the DAD system were 306 and 324 nm for the quantification of *t*-resveratrol and *t*-ε-viniferin respectively.

Gorena et al. [76] (Analysis 3, Table 3) suggested the use of HPLC–DAD for the quantification of stilbenes from grapevine canes of different *Vitis vinifera* varieties. The wavelengths selected in the DAD system were 280 and 306 nm. The identification was performed with an HPLC–DAD–ESI–MS/MS system.

Guerrero et al. [70] (Analysis 4, Table 3) used an HPLC with DAD in full scan mode. The wavelengths used were 280 and 320 nm. This methodology was used by Piñeiro et al. [69] to quantify four stilbenes (*t*-resveratrol, *t*-piceid, piceatannol, and *t*-ε-viniferin) extracted with the microwave-assisted technique.

Billet et al. [68] (Analysis 5, Table 3) suggested HPLC–DAD for the identification and quantification of two stilbenes in grapevine canes from the Pinot noir variety. Quantification was based on pure standards of *t*-resveratrol and *t*-piceatannol with full scans.

The HPLC technique has been used to identify and quantify different stilbenes in grapevine cane extracts. However, their quantification is highly related to the standard preparation, since few of them are commercially available. For this reason, to ensure the correct identification and quantification, HPLC is usually coupled to other analytical techniques such as mass spectrometry (MS) or nuclear magnetic resonance (NMR).

### 4.2. Liquid Chromatography–Mass Spectrometry (LC–MS)

Soural et al. (Analysis 6, Table 3) [66] used HPLC–MS for the identification of *t*-ε-viniferin and r2-viniferin in grapevine canes. To this end, the mass system was equipped with an electrospray (ESI), atmospheric pressure chemical (APCI), and atmospheric pressure photo ionization (APPI) sources and a photodiode array. The APCI capillary temperature was 275 °C, APCI vaporizer temperature 400 °C, sheath gas flow 58 L/min, auxiliary gas flow 10 L/min, source voltage 6 kV, source current 5 µA, and capillary voltage 10 V.

Rodríguez-Cabo et al. [55] (Analysis 7, Table 3) proposed the use of LC separation. The authors injected the samples into a quadrupole time-of-flight mass spectrometry (QToF-MS) system operated at 2 GHz and using HS ([M−H]^−^) mode for the quantification. The LC-QTOF-MS library and previously published data from wine extracts were used for the identification and quantification of the stilbenes.

Sáez et al. [82] (Analysis 8, Table 3) proposed the use of an HPLC system coupled in series to a DAD, an FLD, and a triple-quadrupole mass spectrometer for the identification of stilbenes in grapevine canes. Detection using DAD was performed at 306 and 280 nm, and for FLD the excitation and emission wavelengths were 330 and 374 nm for the stilbenoids. The mass spectrometer used electrospray ionization in negative mode. The source temperature selected was 450 °C, the nebulizer gas pressure was 2.7 bar, and the auxiliary gas pressure 3.4 bar. The *m/z* mass range was set to 100–1200.

Vergara et al. [78] (Analysis 9, Table 3) used HPLC–DAD–ESI–MS/MS for the identification and quantification of stilbenes in grapevine cane extracts from different *Vitis vinifera* varieties. Mass spectrometry was in negative ionization mode, with a drying temperature of 450 °C, ion spray voltage of –4000V, nebulizer gas at 40 psi, and auxiliary gas at 50 psi. The scan range was *m/z* 100–1200.

Ewald et al. [64] (Analysis 10, Table 3) used HPLC–DAD–ESI–MS/MS for the identification of stilbenes in Pinot noir grapevine canes. Mass spectra were recorded in negative ionization mode with a capillary voltage set at 3500 V, the endplate at –500 V, and the capillary exit at −115.0 V. The drying gas was nitrogen at 330 °C and the nebulizer pressure was set to 50 psi, the target mass at *m/z* 400, and the scan range from *m/z* 100 to 3000.

Gorena et al. [76] (Analysis 11, Table 3) selected the HPLC–DAD–ESI–MS/MS system for the identification of stilbenes from grapevine canes from different varieties of *Vitis Vinifera*. Regarding the ESI–MS/MS system, a negative ionization mode was selected with a drying temperature of 450 °C. The nebulizer gas pressure was 40 psi and the auxiliary gas pressure was 50 psi. Finally, the scan range was 100–1200 *m/z*.

Gabaston et al. (Analysis 12, Table 3) [85] selected a UHPLC–DAD/ESI–Q-TOF system. Mass spectrometry analyses were carried out in negative mode, and the drying gas used was nitrogen at 9 L/min at 300 °C with a nebulizer pressure of 25 psi. The sheath gas flow and temperature were set to 11 L/min and 350 °C. The capillary voltage was 4000 V.

### 4.3. Liquid Chromatography–Nuclear Magnetic Resonance (LC–NMR)

NMR is the main technique used for the structural identification of unknown compounds. Coupling with liquid chromatography using different modes allows for the direct analysis of complex extracts [86]. This technique has been successfully applied to grapevine products such as berries [87] and wines [88]. HPLC–NMR was also applied to identify and quantify stilbenes in grapevine cane extracts [74]. Soural et al. [66] used HPLC coupled to a 500 MHz spectrometer equipped with an HCN triple resonance microflow probe to separate and analyze stilbenes from grapevine canes. ^1^H-NMR spectra were collected in on-flow mode. Complete structural elucidation was performed in the stop-flow mode using 2D-NMR spectra. Using this method, *t*-ε-viniferin and r-viniferin were identified in grapevine canes. Similarly, Lambert et al. [74] identified and quantified nine stilbenes and two flavonols using a combination of LC–MS, LC–NMR, and NMR analysis. In this study, seven stilbenes (*t*-piceatannol, *t*-resveratrol, hopeaphenol, isohopeaphenol, *t*-ε-viniferin, ω-viniferin, r-viniferin) were directly identified using LC–NMR in stop-flow mode or by multi-trapping on a FOXY collector connected to the NMR probe before analysis (Analysis 13, Table 3).

In conclusion, grapevine cane extracts were generally analyzed by liquid chromatography coupled to different detectors: UV–Vis, photodiode array (DAD), fluorescence (FLD), and mass and NMR spectroscopies. Initially, the stilbenes were mainly identified by a classical purification and identification process using MS and NMR spectroscopies. UV–Vis or fluorescence detectors were used for quantification, the latter being more specific. Mass spectroscopy allowed the direct identification of the stilbenes in different matrices, with a high sensitivity and specificity in comparison to UV–Vis or fluorescence spectroscopy. Finally, LC–NMR spectrometry was successfully developed for stilbene analysis. Although NMR is less sensitive than mass spectroscopy, it allows the unambiguous identification of each compound, for instance, among isomers.

## 5. Applications

There are a wide variety of methods for extracting stilbenes from grapevine canes. Most of the current bibliography focuses on analyzing and increasing the knowledge about the stilbenes found in grapevine canes. Moreover, some researchers have studied the applications of these extracts based on some of the beneficial properties associated with stilbenes.

### 5.1. Preservative Activity

Preserving wine from both spoiling microorganisms and oxidation is a real challenge for the wine industry. Sulfur dioxide (SO_2_) is the most used preservative in the wine industry due to its antioxidant and antimicrobial properties. It has been shown to inhibit polyphenol oxidase activity during winemaking [89]. Furthermore, due to its antimicrobial activity, SO_2_ controls unwanted microorganism proliferation, which is a key aspect for the wine industry, wherein thousands of liters are produced. However, several human health risks have been associated with SO_2_ use, such as dermatitis, urticaria, angioedema, diarrhea, abdominal pain, bronchoconstriction, and anaphylaxis. SO_2_ is not only used as a preservative in wine, but also in different food matrices, and the amount ingested is accumulated in the organism [90]. The International Organization of Vine and Wine (OIV) has reduced the maximum SO_2_ concentration authorized in wine to 200 mg/L for white and rosé wines, and 150 mg/L for red wines [1].

Consequently, consumer awareness of SO_2_-free products has increased during the last few years [91,92,93]. For this reason, many researchers are currently studying methods that reduce the use of SO_2_ as a preservative while maintaining the quality of the wines. Some chemical compounds have been tested as SO_2_ substitutes: ascorbic acid, thidipropionic acid, sodium hypochlorite, colloidal silver complex, dimethyl dicarbonate, hypophosphorous acid, Trolox C, stannous chloride, and Sporix. Natural products such as lysozyme and bacteriocins have also been studied [94]. Enological tannins combined with lysozyme were also added during the alcoholic fermentation of white wines [95]. Moreover, extracts rich in polyphenols from almond skin and eucalyptus leaves have been evaluated in Verdejo wines during aging in barrels. Good results were found, with no significant changes in the sensory analysis; the extracts did, however, modify the volatile compositions of the wines [96].

Based on the significant properties associated with stilbenes and their antioxidant activity, stilbene-enriched grapevine shoot extracts have been evaluated as alternatives to SO_2_ in winemaking. Grapevine shoot extracts with different stilbene concentrations (29%, 45%, 99%) have been obtained and applied to evaluate their preservative properties. Firstly, studies were developed with Vineatrol (VIN^®^), a commercial grapevine shoot extract with 29% stilbenes. Although VIN was able to preserve wine, its use affected some organoleptic properties, depending on the dose and on the type of wine. In red wines, VIN markers were identified, mainly β-damascenone, guaiacol, *E*-whiskey lactone, and isoeugenol, along with some new odorant zones, astringency, and bitterness. Thus, VIN-treated red wines presented modified organoleptic properties [97,98,99]. In the case of white wines, the results showed the inhibition of malolactic fermentation, methionol preservation, and a decrease in acetaldehyde content. However, after six months of storage in the bottle, some side effects were observed in the VIN wines. The authors concluded that the extract needed to be purified to prevent modifications in both the color and flavor properties of white wines [100]. Consequently, the above results led the authors to look for a grapevine shoot extract with a higher stilbene concentration to prevent undesirable modifications to wine properties. A new grapevine shoot extract with a stilbene richness of 45% was developed. Preliminary assays showed quite similar side effects to those found with the VIN extract (unpublished data). Moreover, Medrano-Padial et al. [101] reported significant damage in two human cell lines when they were exposed to this extract. Consequently, this extract was discarded for further research.

Finally, a pure stilbene extract (99%) from grapevine shoots was developed after some purification steps (unpublished data, under revision). The preliminary assays in wine matrices with the above extract suggest it is a promising tool to control oxidation and spoilage in wines. The side effects have been minimized in wines treated with this extract. Moreover, toxicological studies have been performed. In vitro assays allowed us to propose the pure extract as promising for use in wines (data under revision), although in vivo studies are currently being developed to determine its safety.

In summary, research is developing a new preservative to obtain added-value wine with less SO_2_ (allergic compound) and more bioactive stilbenes.

### 5.2. Antifungal Activity

Today, fungal pathogens in grapevines are a significant challenge to viticulture, resulting in serious consequences such as the loss of many hectares, which entails a critical economic impact. Research into natural alternatives to chemical pesticides is crucial to prevent the main grapevine pathogens such as *Botrytis cinerea* (gray mold), *Plasmopara viticola* (downy mildew), and *Erysiphe necator* (powdery mildew). The production of resveratrol in vine leaves when exposed to pathogens such as *B. cinerea*, *P. viticola*, and *E. necator* was first reported by Langcake and Pryce [102].

Natural extracts rich in stilbenes have received increasing attention due to their antimicrobial activity against grapevine pathogens [103,104,105]. Stilbenes have exhibited significant antifungal activity in vitro and in the field on *P. viticola* and *B. cinerea* [16,106], even in the hemisynthesis of oligomeric stilbenes by oxidative coupling using metals such as copper [16,106,107]. In fact, plants have been found to accumulate the most toxic stilbenes at infection sites [108,109]. Among these compounds, oligomers such as δ-viniferin seem to constitute the most important phytoalexins produced as a response to fungal infection [110]. Pezet and Pont [111] reported that pterostilbene and resveratrol produced in the leaves of *Vitis vinifera* infected by *Botrytis cinerea* acted at the level of membranes. Their concentrations in berries were much less lethal than in vitro concentrations, suggesting that these stilbenes were part of a chemical complex contributing to the natural resistance of grapes. Pezet et al. [112] and Schnee et al. [113] reported that resveratrol, viniferins, and pterostilbene showed significant inhibitory activity on the mobility of *P. viticola* and *E. necator* zoospores and subsequently affected the development of the disease. Finally, Lambert et al. [114] described oligomers (miyabenol C, isohopeaphenol, r-viniferin, and r2-viniferin) as factors reducing the in vitro growth of fungus related to grapevine trunk diseases.

Therefore, based on their significant antifungal activity, stilbenes have been proposed as a natural alternative to the current antifungal treatment. In this sense, grapevine cane extract enriched in stilbenes was suggested for the antifungal treatment of grapevines to prevent the catastrophic consequences of fungal diseases. Richard et al. [115] used Vineatrol against downy mildew in greenhouses and vineyards. At a concentration of 5 g/L, they obtained a comparable effect to that of copper in greenhouses, reducing the frequency of attack by up to 39% and leaf surface infection by up to 61% in vineyards. Furthermore, they demonstrated that stilbene extract acts directly on the fungus, inhibiting the germination and release of zoospores. Recently, a large-scale three-year study evaluated the potential of grapevine shoot extract against natural mildew infection [116]. The 8 g/L extract used presented average reductions in disease incidence and severity comparable to that of 1 g/L of copper. Furthermore, no adverse effects were observed on the grapevine or auxiliary fauna. To increase the efficacy of the extract, they suggest increasing the total stilbene content and using formulations to protect the extract from sun or rainfall.

Gabaston et al. [103] studied the use of different grapevine extracts as natural plant agents to combat *P. viticola,* one of the major fungal pathogens in grapevines. Although all the extracts showed positive results, wood followed by root extract exhibited the highest efficiency. These extracts presented their highest contents in oligomeric stilbenes in comparison with the grapevine cane extract, with the highly lignified parts presenting more oligomeric stilbenes [117]. The oligomeric stilbenes such as r-viniferin, hopeaphenol, and r2-viniferin showed the lowest IC_50_ values against disease development. These values are close to those of pterostilbene and δ-viniferin, the most effective stilbenes previously described against *P. viticola* development [112].

In addition to the direct effect on the pathogen as the contact fungicide, the grapevine shoot extract could act as an elicitor in preventive treatment against *B. cinerea* [118]. The stilbene extract could activate some grapevine defenses, including mitogen-activated protein kinase (MAPK), an early defense response to plant stresses, and the expression of defense-related genes.

Some authors have even suggested the possibility of using artificial inoculation and the micro-analysis of stilbenes to identify and breed grapevines that are resistant to fungal diseases, while still producing wines with good organoleptic profiles [119].

### 5.3. Insecticidal Effects

Gabaston et al. [120] investigated the use of grapevine-root extract with a high stilbene concentration against *Leptinotarsa decemlineata* (Colorado potato beetle). The results showed that the extract inhibited larval development and food intake and showed important toxic activity against the pathogen. Outdoor pot experiments revealed a high efficacy of the extract, with high mortality of *L. decemlineata*, good specificity, and an absence of toxicity against non-targeted organisms. These results were in agreement with those of Sánchez-Gómez et al. [121], who reported the antioxidant, antifeedant, and phytotoxic properties of aqueous extracts of grapevine cane. Similarly, the grapevine cane extract induced chronic mortality to *Spodoptera littoralis* (African cotton leafworm) larvae without antifeedant or acute toxicity effects [122]. The tetramer r-viniferin was proposed as the most active compound.

### 5.4. Viticultural Biostimulants

Biostimulants are defined as “organic extracts obtained from raw materials which contain many different bioactive compounds that are mainly applied on plants by foliar sprays” [123].

In recent years, vegetable extracts have been applied in vineyards to produce alterations in bioactive compound concentrations, resulting in beneficial consequences for grapevines, grapes, or wine quality. For example, Martínez-Gil et al. proposed the use of an aqueous oak extract as a biostimulant in wine. The extract was supplied by Protea France S.A.S., and was obtained by maceration in water at a high temperature. It was applied by Martínez-Gil et al. in Verdejo, Petit Verdot, and Monastrell varieties [124,125,126,127]. Vineyards were treated with different concentrations of the extract. First, the extract was diluted with water to one in four parts and applied (i) once, on the seventh day post-veraison (25%-1 treatment) and (ii) four times (7, 11, 15, and 18 days post-veraison (25%—four treatments). In addition, the undiluted extract was applied once on the seventh day post-veraison (100%—one treatment). Each treatment was applied to 16 plants in the same row, leaving other rows with untreated plants between the different applications to prevent contamination. The leaves of the plants were evenly sprayed with 300 mL of each formulation. An additional row was not treated for a control. The treatments were conducted at 20 °C and the winemaking process was performed under traditional conditions. The results showed a significant influence of the oak extract. Phenolic compounds were assimilated by the grapes and transmitted into their wines, positively impacting the quality of the wine and its chromatic properties, whose evolution over time was more stable.

Thus, grapevine cane extracts can be suggested as biostimulants due to the interesting compounds they contain, such as polyphenols, vitamins, and stilbenes, which can be transferred to grapevines with repercussions in antioxidant, antimicrobial, and antifungal activity. Aqueous extracts from grapevine canes were applied as biostimulants and the results showed an increase in gallic acid, hydroxycinnamoyl tartaric acid, acylated anthocyanins, flavonols, and stilbene content, and a differentiation in the wine quality that could even benefit human health [127,128]. Sánchez-Gomez et al. [129] suggested the use of aqueous extracts from Moscatel grapevine shoots as a biostimulant. Grapevine shoots were sampled four months after the harvesting procedure and left to dry at room temperature for another three months. They had a final humidity of 6.5% (g water/100 g of sample). The influence of the toasting procedure was evaluated as well. To this end, non-toasted samples were ground by a hammer miller, passed through a 10-mesh sieve to get a homogenous sawdust, and kept under vacuum at room temperature. Meanwhile, half of samples were toasted at 180 °C for 45 min. Fifty grams of each one was extracted with 250 mL of boiling water at 100 °C for 15 min and then filtered through a PVDF Durapore 0.45 μm filter. It is important to emphasize that PVDF filters have exhibited an important selectivity to absorb *t*-ε-viniferin from a mixture [130], and thus the composition of the extract may be affected by the filtration step. Airén white grapevines were selected to evaluate the influence of Moscatel grapevine cane extract and the vineyards were treated with 300 mL of each formulation once on the seventh day post-veraison. The resulting wines showed a positive influence of the grapevine shoot extracts, with a reduction in the alcohol content and an increase in volatile and phenolic compounds. Moreover, they were characterized by their fruity and floral aroma.

Toasting grapevine canes has revealed increases in the release and formation of phenolic compounds (including stilbenes) and volatile compounds, and a specific chemical profile consisting of compounds with high added value that are of great interest to the wine sector. This process also induces the degradation of biopolymers that could influence plant absorption [131,132,133].

Cebrián-Tarancón et al. [134,135] suggested the use of toasted grapevine shoot fragments (chips or granules) in the winemaking process. Grapevine shoots from two *Vitis vinifera* cultivars (Airén and Tempranillo Tinto) were pruned 90 days after their grape harvest. The shoots were stored in the dark at room temperature for six months to achieve the highest accumulation of volatile and phenolic compounds. Then, they were toasted in an air circulation oven at 180° for 45 min. Two different formats were tested: 2.5–3.5 cm long chips and granules (2 mm to 2 cm). Those formats were used in Airén and Tempranillo winemaking processes, in which the chips and granules were added in a dose of 12 g/L in two different winemaking steps. In one case, the grapevine shoot chips or granules were added before alcoholic fermentation and removed when this finished. The Airén wines were analyzed at that moment, whereas the Tempranillo Tinto wines were analyzed after the malolactic fermentation. On the other hand, the grapevine cane chips and granules were added after the alcoholic fermentation in the case of the Airén wines, and after malolactic fermentation in the case of the Tempranillo. Maceration was carried out for 35 days. The results showed significant differences according to the presence of grapevine shoot chips or granules in the winemaking process. In general, no significant differences in the main enological parameters were found in either case, but significant differences in the phenolic acid content and stilbene concentration were observed between the treated and control wines, especially when the granulated grapevine shoots were used—those giving a greater contact surface. No significant differences were observed related to the time the grapevine shoots were added. This study confirmed that the addition of grapevine shoots to wines can modulate their chemical compositions when compared with their respective control wines.

The use of grapevine canes as a biostimulant represents an enhanced use of natural extracts that could increase the concentrations of bioactive compounds with positive effects on wine quality.

### 5.5. Health-Related Applications

Stilbenes have been attributed many health-related properties, and therefore the use of stilbene extracts in medical applications has been suggested. Baechler et al. [136] described the topoisomerase inhibiting properties of VIN30, a grapevine shoot extract with a stilbene richness of 30%, and thus its antiproliferative, pro-apoptotic, and antioxidant activities. Neuroprotective effects of VIN30 have also been described in vitro (Biais et al., 2017 [137]; Chaher et al., 2014 [138]; Moreira et al. [139]). Although the low bioavailability of stilbene polymers may be a handicap, a micellar formulation has been proposed as an efficient tool to solve this inconvenience (Calvo-Castro et al., 2018 [140]).

Finally, recent results have been reported about the toxicological effect of an extract from grapevine shoots (with a stilbene richness of 45.4%) in two human cell lines (Medrano-Padial et al., 2019 [101]). The cytotoxicity study showed that the stilbene extract reduced cell viability in both cell lines in the range of concentrations from 30 to 100 µg/mL. In contrast, the extract presented a protective and reductive role against induced oxidative stress. The authors concluded that more research is needed to establish effective and safe concentrations of the stilbene extract.

## 6. Conclusions and Future Prospects

Grapevine canes are an unexploited source of stilbenes. Stilbenes have been shown to be bioactive compounds with important antioxidant, anti-microbial, anti-aging, and even anticancer properties that may benefit human health. Therefore, the possibility of obtaining stilbene extracts from grapevine canes, usually considered a waste product, is a great opportunity for the wine industry.

Many studies have determined the stilbene compositions in grapevine canes, especially in the *Vitis vinifera* genus. Up to 41 stilbenes have been identified in grapevine canes, although huge variability in both their identification and quantification has been observed. This variability could be (i) intrinsic, since the variety, climate conditions, soil, and grapevine management affect stilbene content, and/or (ii) extrinsic, resulting from extraction and analysis methods. The Pinot noir variety is one of the highest stilbene producers, while stressful conditions are recommended to increase stilbene oligomers. Regarding extraction, solid–liquid extraction with ethanol:water or acetone is the most used. For identification, HPLC–MS coupled to NMR is the most powerful technique, whereas for quantification, HPLC–DAD is the most used.

The state-of-the-art from all the above has been quite well-described. The challenge now is to discover applications for stilbene extracts (from grapevine cane) to take full advantage of all the above knowledge. Five important applications are currently being studied: (a) as an alternative to SO_2_ in winemaking; (b) as antifungal treatments in grapevines; (c) as insecticides; (d) as biostimulants in grapevines; and (e) for health-related applications. These applications imply major innovations in viticulture and enology, and a new alternative use for this waste. However, considering the properties of stilbenes, cosmetic, pharmaceutical, or nutraceutical applications may be a possibility. In fact, stilbene extracts from other sources are already being used in these fields. Thus, the conclusion can be drawn that grapevine cane extracts could be a promising breakthrough in the fields of viticulture and enology thanks to the range of prospects they offer. Finally, it is worth mentioning that all the above contributes to both the sustainability and circular bioeconomy in the vineyard.

## Figures and Tables

**Figure 1 biomolecules-10-01195-f001:**
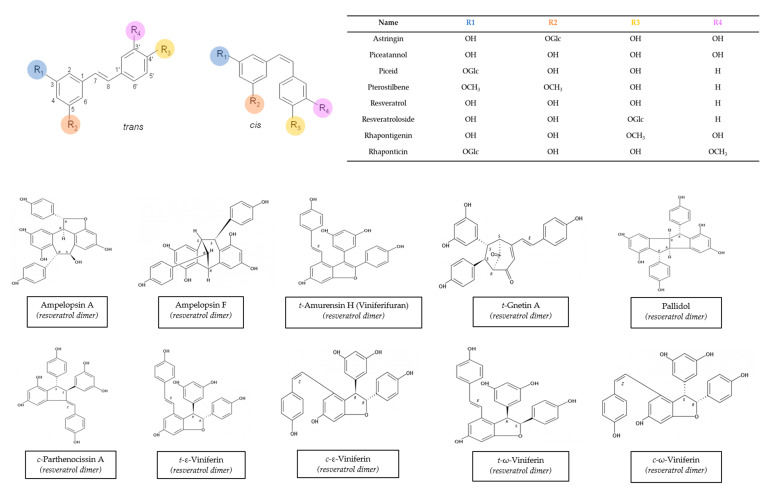
Structures of main stilbenes from grapevine canes. Structures obtained from the ISVV-Polyphenols reference database [49].

**Figure 2 biomolecules-10-01195-f002:**
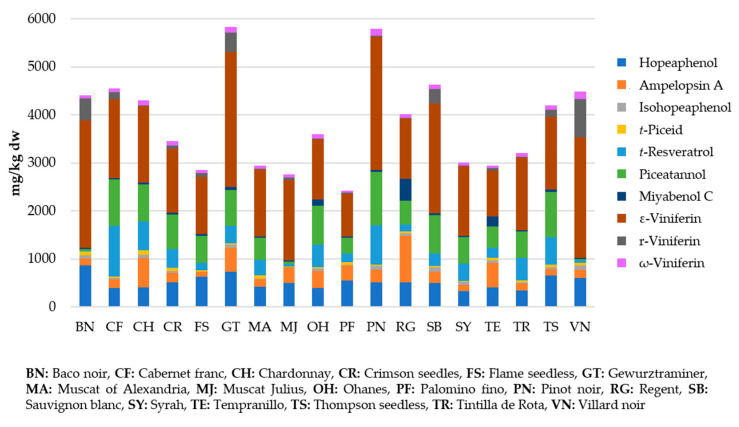
Stilbene concentrations in grapevine canes from different varieties of *Vitis vinifera* (Adapted from [70]).

**Table 1 biomolecules-10-01195-t001:** Stilbenes described in grapevine canes from different *Vitis* species [23,54,55,56,57,58,59,60,61,62,63,64,65,66,67,68,69,70].

Stilbene	Species Concentration (mg/kg dw)																	
	*V. amurensis*	*V. arizonica*	*V. berlandieri*	*V. betulifolia*	*V. X champinii*	*V. cinerea*	*V. coignetiae*	*V. davidii*	*V. X doaniana*	*V. heyneana*	*V. flexuosa*	*V. labrusca*	*V. mustangensis*	*V. pentagona*	*V. riparia*	*V. ruperstris*	*V. vinifera*	*V. thunbergii*
*t-/c*-Astringin	nd	nd	nd	nd	nd	nd	367.0 ± 17.0	nd	nd	Nd	nd	nd	nd	nd	nd	nd	nd	nd
*t*-Piceatannol	1962.2 ± 122.4	1100.8 ± 21.5	894.4 ± 6.8	194.1 ± 13.6	599.7 ± 71.8	1195.1 ± 73.6	nq	nd	1151.1 ± 196.5	Nd	nd	377.0 ± 34.3	199.6 ± 34.9	nd	615.4 ± 18.1	1182.7 ± 26.7	1710.0 ± 4.0	nd
*t-c*-Piceid	nq	nd	64.2 ± 2.5	101.0 ± 1.5	nd	133.8 ± 8.6	201.0 ± 25.0	nd	nd	Nd	nd	nd	nd	nd	291.7 ± 11.6	257.0 ± 9.6	284.0 ± 36.0	nd
Pterostilbene	nd	nd	nd	nd	nd	nd	547.0 ± 124.0	nd	nd	Nd	nd	nd	nd	nd	nd	nd	nd	nd
*t-/c*-Resveratrol	5432.9 ± 208.6	2412.4 ± 7.5	1951.2 ± 86.0	191.0 ± 2.4	2534.9 ± 15.0	3165.7 ± 131.3	25.0 ± 4.0	1049.0 ± 138.0	3572.2 ± 76.7	Nd	nd	1028.2 ± 53.0	364.0 ± 2.1	839.0 ± 31.0	1666.1 ± 9.4	3966.5 ± 52.5	66200.0 ±11.0	nq
*t-/c*-Resveratroloside	213.6 ± 7.2	nd	nd	nd	nd	nd	nd	nd	nd	Nd	nd	nd	nd	nd	nd	nd	nd	nd
*t*-Rhapontigenin	nd	nd	nd	nd	nd	nd	22.0 ± 3.0	nd	nd	Nd	nd	nd	nd	nd	nd	nd	nd	nd
Rhaponticin	nd	nd	nd	nd	nd	nd	52.0 ± 6.0	nd	nd	Nd	nd	nd	nd	nd	nd	nd	nd	nd
*t*-Ampelopsin A	nq	nd	nd	149.6 ± 6.6	nd	nd	nd	nd	nd	Nq	nd	nd	nd	nd	nd	nd	220.0 ± 16.0	nd
Ampelopsin F	nq	nd	nd	nd	nd	nd	nd	nd	nd	Nd	nd	nd	nd	nd	nd	nd	360.0 – nq	nq
*t*-Amurensin H	nd	nd	nd	nd	nd	nd	nd	nd	nd	Nd	nd	nd	nd	nd	nd	nd	nd	nq
*t*-Gnetin A	nd	nd	nd	nd	nd	nd	nd	nd	nd	Nd	nq	nd	nd	nd	nd	nd	nd	nd
Pallidol	nd	nd	nd	nd	nd	nd	nd	nd	nd	Nd	nd	nd	nd	nd	nd	nd	80.0 – nq	nd
*c*-Parthenocissin A	nd	nd	nd	nd	nd	nd	nd	nd	nd	Nd	nd	nd	nd	nd	nd	nd	220.0 – nq	nd
*t-/c*-ε-Viniferin	4510.4 ± 20.9	3715.5 ± 11.5	1264.5 ± 33.8	1397.4 ± 16.9	4054.9 ± 87.5	1610.9 ± 29.0	nd	nd	4352.5 ± 35.8	Nq	nd	4683.7 ± 99.7	3254.3 ± 68.5	nd	5739.0 ± 45.8	3912.6 ± 301.7	40600.0 ± 47.0	nd
*t-/c*-ω-Viniferin	nd	122.2 ± 3.1	nd	65.1 ± 2.6	365.9 ± 9.3	nd	nd	nd	288.3 ± 3.2	Nd	nd	214.2 ± 3.2	270.5 ± 6.4	nd	156.8 ± 3.2	374.3 ± 26.9	85.2 ± 40.0	nd
Vitisinol A	nd	nd	nd	nd	nd	nd	nd	nd	nd	nd	nd	nd	nd	nd	nd	nd	nd	nq
Vitisinol B	nd	nd	nd	nd	nd	nd	nd	nd	nd	nd	nd	nd	nd	nd	nd	nd	nd	nq
*t*-Vitisinol C	nd	nd	nd	nd	nd	nd	nd	nd	nd	nd	nd	nd	nd	nd	nd	nd	nd	nq
Vitisinol D	nd	nd	nd	nd	nd	nd	nd	nd	nd	nd	nd	nd	nd	nd	nd	nd	nd	nq
*t*-Vitisinol E	nd	nd	nd	nd	nd	nd	nd	nd	nd	nd	nd	nd	nd	nd	nd	nd	nd	nq
Vitisinol G	nd	nd	nd	nd	nd	nd	nd	nd	nd	nd	nd	nd	nd	nd	nd	nd	nd	nq
*t*-Gnetin H	nq	nd	nd	nd	nd	nd	nd	nd	nd	nd	nd	nd	nd	nd	nd	nd	nd	nd
α-Viniferin	nd	nd	nd	nd	nd	nd	nd	nd	nd	nd	nd	nd	nd	nd	nd	nd	105.6 ± 98.4	nd
*t*-Amurensin B	567.1 ± 14.7	nd	nd	55.9 ± 2.5	nd	nd	nd	nd	nd	nd	nd	nd	nd	nd	nd	nd	nd	nd
Amurensin G	nq	nd	nd	nd	nd	nd	nd	nd	nd	nd	nd	nd	nd	nd	nd	nd	nd	nd
Ampelopsin C	nd	nd	nd	nd	nd	nd	nd	nd	nd	nq	nd	nd	nd	nd	nd	nd	nd	nq
*t-/c*-Ampelopsin E	1615.1 ± 35.4	428.4 ± 3.5	nd	nd	nd	nd	nd	nd	nd	nd	nd	nd	nd	nd	575.6 ± 16.9	nd	nd	nq
*t-/c*-Miyabenol C	nd	nd	nd	nd	nd	nd	nd	nd	nd	nd	nd	nd	nd	nd	nd	nd	1060.0 –nq	nd
*t*-Vitisinol F	nd	nd	nd	nd	nd	nd	nd	nd	nd	nd	nd	nd	nd	nd	nd	nd	nd	nq
Ampelopsin H	nd	nd	nd	nd	nd	nd	nd	nd	nd	nd	nd	nd	nd	nd	nd	nd	40.0 – nq	nd
Flexuosol A	nd	nd	Nd	nd	nd	nd	nd	nd	nd	nd	nq	nd	nd	nd	nd	nd	nd	nd
Heyneanol A	nd	nd	Nd	Nd	nd	nd	nd	nd	nd	nq	nd	nd	nd	nd	nd	nd	nd	nd
Hopeaphenol	nd	nd	Nd	Nd	nd	nd	nd	nd	nd	nd	nq	nd	nd	nd	nd	nd	1468.2 – nq	nd
Isohopeaphenol	nd	nd	nd	nd	nd	nd	nd	nd	nd	nd	nd	nd	nd	nd	nd	nd	120.0 – nq	nd
Miyabenol A	nd	nd	nd	nd	nd	nd	nd	nd	nd	nd	nd	nd	nd	nd	nd	nd	nd	nq
r2-Viniferin	nq	nd	nd	nd	nd	nd	nd	nd	nd	nd	nq	nd	nd	nd	nd	nd	15200.5 ± 60.0	nq
r-Viniferin	972.1 ± 48.9	4279.6 ± 62.5	2038.0 ± 98.1	nd	5031.8 ± 95.2	2531.7 ± 69.4	nd	nd	2506.2 ± 12.0	nd	nd	5051.0 ± 237.7	6966.2 ± 69.2	nd	1950.7 ± 24.8	4916.1 ± 412.3	2159.0 – nq	nq
*t-/c*-Vitisin B	nd	nd	nd	nd	nd	nd	nq	nd	nd	nd	nd	nd	nd	nd	nd	nd	nq	nq
Vitisin C	nd	nd	nd	nd	nd	nd	nd	nd	nd	nd	nd	nd	nd	nd	nd	nd	nd	nq
Viniferal	nd	nd	nd	nd	nd	nd	nd	nd	nd	nd	nd	nd	nd	nd	nd	nd	nd	nq
Viniferol E	nd	nd	nd	nd	nd	nd	nd	nd	nd	nd	nd	nd	nd	nd	nd	nd	140.0 – nq	nd

nd, not detected; nq, not quantified; *t-, trans*-; *c-, cis-.*

**Table 2 biomolecules-10-01195-t002:** Summary of conditions of the methods for extracting stilbenes from grapevine cane.

Extraction Technique	Method	Conditions	Concentration (mg/Kg dw)	References
Solid-Liquid Extraction	1	Ethanol:H_2_O (80:20 *v*/*v*) Ultrasonic homogenizer sonification. 4 Times with renovation of solvent. Centrifugation	Cabernet sauvignon: *t*-Resveratrol (1639 ± 15) and *t*-ε-Viniferin (2203 ± 29) Merlot: *t*-Resveratrol (2409 ± 103) and *t*-ε-Viniferin (1656 ± 355) Regent: *t*-Resveratrol (753 ± 27) and *t*-ε-Viniferin (1218 ± 40) Riesling: *t*-Resveratrol (1994 ± 34) and *t*-ε-Viniferin (1928 ± 96) Pinot gris: *t*-Resveratrol (1941 ± 32) and *t*-ε-Viniferin (3297 ± 70) Sauvignon blanc: *t*-Resveratrol (2010 ± 124) and *t*-ε-Viniferin (3329 ± 296) Pinot noir: *t*-Resveratrol (1908 ± 124) and *t*-ε-Viniferin (2790 ± 123) Pinot blanc: *t*-Resveratrol (3199 ± 95) and *t*-ε-Viniferin (2125 ± 155)	Vergara et al. [78]; Ewald et al. [64]
	2	Ethanol:H_2_O (80:20 *v*/*v*) Ultrasonic 4 times with renovation of solvent. Centrifugation	Pinot noir: Ampelopsin A (204 ± 1), *t*-Piceatannol (374 ± 5), *t*-Resveratrol (3655 ± 4), Hopeaphenol (60 ± 9), *t*-ε-Viniferin (1445 ± 1) and r-Viniferin (traces) Gewürztraminer: *t*-Piceatannol (233 ± 18), *t*-Resveratrol (3599 ± 116), Hopeaphenol (65 ± 6), *t*-ε-Viniferin (542 ± 29) and r-Viniferin (49 ± 1) Tinta pais: *t*-Piceatannol (192 ± 9), *t*-Resveratrol (3034 ± 58), Hopeaphenol (69 ± 8), *t*-ε-Viniferin (114 ± 4) and r-Viniferin (traces) Cabernet sauvignon: *t*-Piceatannol (283 ± 8), *t*-Resveratrol (2407 ± 110), Hopeaphenol (79 ± 14), *t*-ε-Viniferin (333 ± 10) and r-viniferin (traces) Syrah: *t*-Piceatannol (261 ± 14), *t*-Resveratrol (3591 ± 188), Hopeaphenol (traces), *t*-ε-Viniferin (368 ± 14) and r-viniferin (traces) Carmenère: *t*-Piceatannol (212 ± 23), *t*-Resveratrol (2811 ± 98), Hopeaphenol (traces), *t*-ε-Viniferin (324 ± 7) and r-viniferin (traces) Sauvignon blanc: *t*-Piceatannol (182 ± 14), *t*-Resveratrol (136 ± 20), Hopeaphenol (74 ± 6), *t*-ε-Viniferin (508 ± 44) and r-viniferin (traces) Garnacha Tintorera: *t*-Piceatannol (308 ± 12), *t*-Resveratrol (4074 ± 125), Hopeaphenol (59 ± 10), *t*-ε-Viniferin (512 ± 2) and r-viniferin (traces) Cinsault: *t*-Piceatannol (143 ± 1), *t*-Resveratrol (1506 ± 48), Hopeaphenol (traces), *t*-ε-Viniferin (324 ± 10) and r-Viniferin (traces) Moscatel de Alejandria: *t*-Piceatannol (288 ± 22), *t*-Resveratrol (4941 ± 128), Hopeaphenol (traces), *t*-ε-Viniferin (343 ± 23) and r-viniferin (traces) Semillon: *t*-Piceatannol (216 ± 5), *t*-Resveratrol (2112 ± 18), Hopeaphenol (traces), *t*-ε-Viniferin (621 ± 13) and r-viniferin (traces) Merlot: *t*-Piceatannol (170 ± 8), *t*-Resveratrol (1936 ± 65), Hopeaphenol (32 ± 3), *t*-ε-Viniferin (294 ± 12) and r-viniferin (traces).	Gorena et al. [76]; Sáez et al. [82]
	3	Acetone:H_2_O (60:40 *v*/*v*) Centrifugation Dryness Resolved methanol/water (1:1 *v*/*v*)	Cabernet sauvignon: *t *-Piceatannol (735 ± 142), *t*-Resveratrol (871 ± 202), *t*-ε-Viniferin (2379 ± 1123), r-Viniferin (420 ± 109) and Miyabenol C (30 ± 12) Carignan: *t*-Piceatannol (519 ± 80), *t*-Resveratrol (880 ± 304), Hopeaphenol (1439 ± 214), *t*-ε-Viniferin (967 ± 71), r-Viniferin (traces) and Miyabenol C (87 ± 21) Chardonnay: *t*-Piceatannol (190 ± 67), *t*-Resveratrol (190 ± 87), Hopeaphenol (766 ± 149), *t*-ε-Viniferin (2089 ± 334), r-Viniferin (traces) and Miyabenol C (traces) Chenin: *t*-Piceatannol (1227 ± 267), *t*-Resveratrol (794 ± 161), Hopeaphenol (623 ± 175), *t*-ε-Viniferin (2218 ± 274), r-viniferin (traces) and Miyabenol C (traces) Cinsault: *t*-Piceatannol (298 ± 268), *t*-Resveratrol (486 ± 226), Hopeaphenol (339 ± 96), *t*-ε-Viniferin (1629 ± 100), r-Viniferin (traces) and Miyabenol C (1060 ± 12) Gamay: *t*-Piceatannol (843 ± 138), *t*-Resveratrol (980 ± 201), Hopeaphenol (1085 ± 182), *t*-ε-Viniferin (1828 ± 157), r-Viniferin (102 ± 53) and Miyabenol C (traces) Gewürztraminer: *t*-Piceatannol (490 ± 150), *t*-Resveratrol (649 ± 290), Hopeaphenol (1118 ± 357), *t*-ε-Viniferin (2199 ± 379), r-Viniferin (1116 ± 380) and Miyabenol C (traces) Grenache: *t*-Piceatannol (372 ± 195), *t*-Resveratrol (752 ± 392), Hopeaphenol (465 ± 123), *t*-ε-Viniferin (1792 ± 110), r-Viniferin (88 ± 22) and Miyabenol C (traces) Melon: *t*-Piceatannol (561 ± 359), *t*-Resveratrol (963 ± 189), Hopeaphenol (645 ± 188), *t*-ε-Viniferin (1970 ± 193), r-Viniferin (126 ± 44) and Miyabenol C (traces) Merlot: *t*-Piceatannol (947 ± 353), *t*-Resveratrol (1181 ± 189), Hopeaphenol (642 ± 163), *t*-ε-Viniferin (2263 ± 220), r-Viniferin (146 ± 48) and Miyabenol C (22 ± 14) Pinot noir: *t*-Piceatannol (1710 ± 224), *t*-Resveratrol (1526 ± 293), Hopeaphenol (1126 ± 294), *t*-ε-Viniferin (3737 ± 421), r-Viniferin (313 ± 156) and Miyabenol C (73 ± 22) Riesling: *t*-Piceatannol (270 ± 101), *t*-Resveratrol (605 ± 258), Hopeaphenol (1468 ± 601), *t*-ε-Viniferin (1716 ± 411), r-Viniferin (88 ± 54) and Miyabenol C (174 ± 12) Sauvignon blanc: *t*-Piceatannol (607 ± 294), *t*-Resveratrol (730 ± 34), Hopeaphenol (841 ± 263), *t*-ε-Viniferin (2697 ± 167), r-Viniferin (369 ± 212) and Miyabenol C (36 ± 17) Semillon: *t*-Piceatannol (471 ± 208), *t*-Resveratrol (872 ± 263), Hopeaphenol (287 ± 124), *t*-ε-Viniferin (2448 ± 186), r-Viniferin (252 ± 106) and Miyabenol C (traces) Syrah: *t*-Piceatannol (460 ± 253), *t*-Resveratrol (481 ± 373), Hopeaphenol (586 ± 456), *t*-ε-Viniferin (2507 ± 462), r-Viniferin (182 ± 244) and Miyabenol C (38 ± 14) Ugni blanc: *t*-Piceatannol (1056 ± 295), *t*-Resveratrol (689 ± 1190), Hopeaphenol (818 ± 202), *t*-ε-Viniferin (2292 ± 259), r-Viniferin (138 ± 21) and Miyabenol C (39 ± 12)	Lambert et al. [74]; Guerrero et al. [70]
High-Pressure Methods	4	Sample in column heated and pressurized Ethanol in dynamic mode. 3 Times folds	Cabernet moravia: *t*-Resveratrol (46500), *t*-ε-Viniferin (7300) and r2-Viniferin (500)	Zachová et al. [67]
	4	Sample in column heated and pressurized. Solvent with 3 cycles Supernatant evaporated until dry extract	Cabernet moravia (Acetone): *t*-resveratrol (66200), *t*-ε-Viniferin (40600) and r2-Viniferin (15200) Cabernet moravia (mix-distilled alcohol:H_2_O): *t*-Piceid (36 ± 6), *t*-Piceatannol (195 ± 3), *t*-Resveratrol (1215 ± 161) and *t*-ε-Viniferin (2141 ± 14)	Zachová et al. [67]; Rodríguez-Cabo et al. [55]
Microwave-assisted extraction	5	Sample at 125 °C Ethanol:H_2_O (80:20 *v*/*v*)	Cabernet Franc: Piceatannol (28 ± 6), *t*-Resveratrol (27 ± 2) and *t*-ε-Viniferin (222 ± 10) Cabernet Sauvignon: *t*-Resveratrol (533 ± 21) and *t*-ε-Viniferin (1109 ± 16) Carmenére: Piceatannol (84 ± 6), *t*-Resveratrol (77 ± 4) and *t*-ε-Viniferin (178 ± 6) Chardonnay: Piceatannol (19 ± 1), *t*-Resveratrol (37 ± 3) and *t*-ε-Viniferin (692 ± 6) Gewürtztraminer: Piceatannol (494 ± 24), *t*-Resveratrol (52361 ± 107) and *t*-ε-Viniferin (2567 ± 24) Jaen tinto: Piceatannol (66 ± 1), *t*-Resveratrol (402 ± 17) and *t*-ε-Viniferin (457 ± 7) Malbec: *t*-Piceid (228 ± 10), Piceatannol (66 ± 9), *t*-Resveratrol (960 ± 29) and *t*-ε-Viniferin (1414 ± 17) Marselan: Piceatannol (40 ± 1), *t*-Resveratrol (11 ± 1) and *t*-ε-Viniferin (201 ± 4) Melonera: *t*-Piceid (241 ± 33), Piceatannol (59 ± 1), *t*-Resveratrol (427 ± 18) and *t*-ε-Viniferin (1102 ± 12) Merlot: Piceatannol (57 ± 2), *t*-Resveratrol (69 ± 1) and *t*-ε-Viniferin (1017 ± 14) Moscatel Alejandria: Piceatannol (29 ± 1), *t*-Resveratrol (86 ± 2) and *t*-ε-Viniferin (1922 ± 27) Moscatel julius: Piceatannol (111 ± 11), *t*-Resveratrol (24 ± 2) and *t*-ε-Viniferin (63 ± 1) Palomino fino: *t*-Piceid (167 ± 5), Piceatannol (21 ± 0), *t*-Resveratrol (142 ± 17) and *t*-ε-Viniferin (870 ± 12) Palomino negro: *t*-Resveratrol (498 ± 10) and *t*-ε-Viniferin (816 ± 8) Petit verdot: Piceatannol (77 ± 3), *t*-Resveratrol (20 ± 0) and *t*-ε-Viniferin (47 ± 0) Pinot noir: *t*-Resveratrol (640 ± 24) and *t*-ε-Viniferin (3543 ± 70) Regent: Piceatannol (82 ± 3), *t*-Resveratrol (51 ± 0) and *t*-ε-Viniferin (67 ± 3) Sauvignon blanc: *t*-Resveratrol (519 ± 35) and *t*-ε-Viniferin (3100 ± 65) Syrah: Piceatannol (55 ± 3), *t*-Resveratrol (469 ± 18) and *t*-ε-Viniferin (1031 ± 17) Tannat: *t*-Piceid (284 ± 15), Piceatannol (4 ± 0), *t*-Resveratrol (460 ± 6) and *t*-ε-Viniferin (1032 ± 8) Tempranillo: *t*-Piceid (76 ± 2), Piceatannol (43 ± 6), *t*-Resveratrol (204 ± 8) and *t*-ε-Viniferin (602 ± 6) Tintilla de Rota: Piceatannol (45 ± 2), *t*-Resveratrol (486 ± 21) and *t*-ε-Viniferin (1275 ± 15) Vijiriega: Piceatannol (160 ± 6), *t*-Resveratrol (1529 ± 44) and *t*-ε-Viniferin (620 ± 8) Zinfandel: *t*-Piceid (192 ± 1), *t*-Resveratrol (243 ± 5) and *t*-ε-Viniferin (1080 ± 9)	Piñeiro et al. [69]
Subcritical Water	6	Sample in H_2_O at 160 °C and under pressure	Merlot: Piceid (70), Piceatannol (130), Resveratrol (650), Ampelopsin A (220), Ampelopsin F (360), Pallidol (80), Parthenocissin A (200), ε-Viniferin (300), w-Viniferin (40), Viniferol E (140), Hopeaphenol (340), Isohopeaphenol (120), Ampelopsin H (40) and r2-Viniferin (60)	Gabaston et al. [85]

**Table 3 biomolecules-10-01195-t003:** Analytical techniques used to identify stilbenes in grapevine cane samples.

	Analysis	Identified Compounds from Grapevine Cane	References
HPLC–DAD/FLD	1	*t*-Resveratrol, *t*-ε-Viniferin and r2-Viniferin	[66,67]
	2	*t*-Resveratrol and *t*-ε-Viniferin	[64]
	3	*t*-Resveratrol (306 nm), *t*-Piceid (304 nm), *t*-Piceatannol (323 nm), Ampelopsin A (208 nm), Hopeaphenol (282 nm), r-Viniferin (326 nm), r2-viniferin (328 nm) and *t*-ε-Viniferin (323 nm)	[76]
	4	Hopeaphenol, Isohopeaphenol and Ampelopsin A (280 nm); *t*-Resveratrol (306 nm); *t*-Piceid, Piceatannol, *t*-ε-Viniferin, r-Viniferin and *t*-ω-Viniferin (320 nm)	[69,70]
	5	*t*-Resveratrol and *t*-Piceatannol	[68]
LC–MS	6	*t*-ε-Viniferin (C_28_H_22_O_6_): C_28_H_22_O_6_^+^ (*m/z* 455.1482), C_28_H_21_O_5_^+^(*m/z* 437.1373), C_22_H_17_O_5_^+^ (*m/z* 361.0740), and C_13_H_11_O_3_^+^ (*m/z* 215.0709) and r2-viniferin (C_56_H_42_O_12_): C_56_H_42_O_12_^+^ (*m/z* 907.2745), C_35_H_27_O_7_^+^ (*m/z* 559.1709), C_28_H_21_O_6_^+^ (*m/z* 453.1339), C_22_H_17_O_5_^+^ (*m/z* 361.1038), and C_13_H_11_O_3_^+^ (*m/z* 215.0690)	[66]
	7	α-Viniferin C_24_H_30_O_9 (_*m/z* 677.1812), Resveratrol trimer A C_42_H_32_O_9 (_*m/z* 679.1964), Resveratrol trimer B C_42_H_32_O_9 (_*m/z* 679.1968), Resveratrol tetramer A C_56_H_42_O_12 (_*m/z* 905.2598) and Resveratrol tetramer B C_56_H_42_O_12 (_*m/z* 905.2612).	[55]
HPLC–DAD-FLD–MS/MS	8	*t*-Piceid (DAD λ_max_ = 304 - 315 nm, FLD λ_Exc-Emis_ = 330 - 374 nm, *m/z* 389), Ampelopsin A (DAD λ_max_ = 280 nm, FLD λ_Exc-Emis_ = 230 - 320 nm, *m/z* 469), *t*-Piceatannol (DAD λ_max_ = 324 nm, FLD λ_Exc-Emis_ = 330 - 374 nm, *m/z* 243), Pallidol (DAD λ_max_ = 280 nm, *m/z* 253), *t*-Resveratrol (DAD λ_max_ = 306 nm, FLD λ_Exc-Emis_ = 330 - 374 nm, *m/z* 227), Hopeaphenol (DAD λ_max_ = 280 nm, FLD λ_Exc-Emis_ = 230 - 320 nm, *m/z* 905), *t*-ε-Viniferin (DAD λ_max_ = 324 nm, FLD λ_Exc-Emis_ = 330 - 374 nm, *m/z* 453), *t*-δ-Viniferin (DAD λ_max_ = 324 nm, FLD λ_Exc-Emis_ = 330 - 374 nm, *m/z* 453), *t*-ω-Viniferin (DAD λ_max_ = 324 nm, FLD λ_Exc-Emis_ = 330 - 374 nm, *m/z* 453) and r-Viniferin (DAD λ_max_ = 326 nm, FLD λ_Exc-Emis_ = 330 - 374 nm, *m/z* 905).	[82]
HPLC–DAD–ESI–MS/MS	9	*t*-Piceid (λ_max_ = 304 - 315 nm, *m/z* 389), Ampelopsin A (λ_max_ = 280 nm, *m/z* 469), *t*-Piceatannol (λ_max_ = 323 - 303 nm, *m/z* 243), *t*-Resveratrol (λ_max_ = 304 - 316 nm, *m/z* 227) and *t*-ε-Viniferin (λ_max_ = 308 - 322 nm, *m/z* 453)	[78]
	10	Ampelopsin A (*m/z* 469), Hopeaphenol (*m/z* 905), Piceatannol (*m/z* 243), Resveratrol (*m/z* 227), r2-Viniferin (*m/z* 905), Miyabenol C (*m/z* 679), *t*-ε-Viniferin (*m/z* 453) and r-Viniferin (*m/z* 905)	[64]
	11	*t*-Resveratrol (*m/z* 227), *t*-Piceid (*m/z* 389), *t*-Piceatannol (*m/z* 243), Ampelopsin A (*m/z* 469), Hopeaphenol (*m/z* 906), r-Viniferin (*m/z* 906), r2-viniferin (*m/z* 906) and *t*-ε-Viniferin (*m/z* 453)	[76]
UHPLC–DAD/ESI–Q-TOF	12	*t*-Resveratrol, *t*-Piceid, *t*-Piceatannol, Ampelopsin A, Ampelopsin F, Pallidol, *t*-Parthenocissin A, Miyabenol C, Ampelopsin E, Viniferol E, Ampelopsin H, Hopeaphenol, Isohopeaphenol, r-Viniferin, r2-viniferin, *t*-ω-Viniferin, and *t*-ε-Viniferin	[85]
HPLC–NMR	13	*t*-Piceatannol: ^1^H-NMR δ (ppm) 7.00 (1H, d, *J* = 2.0 Hz, H-2), 6.93 (1H, d, *J* = 16.4 Hz, H-7), 6.88 (1H, dd, *J* = 2.0, 8.4 Hz, H-6) 6.80 (1H, d, *J* = 16.4 Hz, H-8), 6.77 (1H, d, *J* = 8.4, H-5), 6.45 (2H, d, *J* = 2.1 Hz, H-10,14), 6.14 (1H, t, *J* = 2.1 Hz, H-12) *t*-Resveratrol: ^1^H-NMR δ (ppm) 7.36 (2H, d, *J* = 8.5 Hz, H-2,6), 6.99 (1H, d, *J* = 16.4 Hz, H-7), 6.82 (1H, d, *J* = 16.4 Hz, H-8), 6.76 (2H, d, *J* = 8.5, H-3,5), 6.44 (2H, d, *J* = 2.1 Hz, H-10,14), 6.13 (1H, t, *J* = 2.1 Hz, H-12) Hopeaphenol: ^1^H-NMR δ (ppm) 7.07 (2H, d, *J* = 8.5 Hz, H-2b,6b), 6.79 (2H, d, *J* = 8.5 Hz, H-2a,6a), 6.76 (2H, d, *J* = 8.5 Hz, H-3b,5b), 6.56 (2H, d, *J* = 8.5 Hz, H-3a,5a), 6.39 (1H, brs, H-12b), 6.19 (1H, brs, H-14b), 5.73 (1H, d, *J* = 12.2 Hz, H-7b), 5.72 (1H, brs, H-12a), 5.42 (1H, d, brs, H-14a), 4.85 (1H, brs, H-7a), 4.08 (1H, d, *J* = 12.2 Hz, H-8b), 3.76 (1H, brs, H-8a) Isohopeaphenol: ^1^H-NMR δ (ppm) 7.46 (2H, d, *J* = 8.4 Hz, H-2a,6a), 6.95 (2H, d, *J* = 8.4 Hz, H-3a,5a), 6.30 (2H, d, *J* = 8.4 Hz, H-2b,6b), 6.23 (2H, d, J = 8.4, H-3b,5b), 6.22 (1H, brs, H-12a), 6.01 (1H, brs, H-14a), 5.80 (1H, d, brs, H-12b), 5.51 (1H, d, J = 10.8 Hz, H-7a), 5.31 (1H, d, brs, H-14b), 5.27 (1H, d, J = 10.8 Hz, H-8a), 4.77 (1H, brs, H-7b), 3.23 (1H, brs, H-8b) *t*-ε-Viniferin: ^1^H-NMR δ (ppm) 7.14 (2H, d, *J* = 8.3 Hz, H-2a,6a), 7.11 (2H, d, *J* = 8.3 Hz, H-2b,6b), 6.87 (1H, d, *J* = 16.4, H-7b), 6.76 (2H, d, *J* = 8.5 Hz, H-3a,5a), 6.69 (2H, d, *J* = 8.8 Hz, H-3b,5b), 6.62 (1H, d, *J* = 1.8, H-14b), 6.59 (1H, d, *J* = 16.4 Hz, H-8b), 6.28 (1H, d, *J* = 1.8 Hz, H-12b), 6.12 (2H, d, *J* = 2.01 Hz, H-10a,14a), 6.09 (1H, t, *J* = 2.1 Hz, H-12a), 5.39 (1H, d, *J* = 5.7 Hz, H-7a), 4.45 (1H, d, *J* = 5.7 Hz, H-8a) *t*-ω-Viniferin: ^1^H-NMR δ (ppm) 7.14 (2H, d, *J* = 8.4 Hz, H-2b,6b), 6.95 (2H, d, *J* = 8.4 Hz, H 2a,6a), 6.90 (1H, d, *J* = 16.4 Hz, H-7b), 6.69 (2H, d, *J* = 8.4 Hz, H-3b,5b), 6.63 (1H, brs, H-14b), 6.61 (1H, d, *J* = 16.4 Hz, H-8b), 6.55 (2H, d, *J* = 8.4 Hz, H-3a,5a), 6.32 (1H, d, brs, H-12b), 5.84 (1H, d, *J* = 8.5 Hz, H-7a), 5.83 (1H, brs, H-12a), 5.70 (2H, brs, H-10a,14a), 4.64 (1H, d, *J* = 8.5 Hz, H-8a) r-Viniferin: ^1^H-NMR δ (ppm) 7.15 (2H, d, *J* = 8.5 Hz, H-2a,6a), 7.13 (2H, d, *J* = 8.5 Hz, H-2d,6d), 7.05 (1H, brd, *J* = 8.2 Hz,-H-6b), 6.79 (2H, d, *J* = 8.4 Hz, H-3a,5a),6.77 (2H, d, *J* = 8.4 Hz, H-3d,5d), 6.73 (1H, d, *J* = 16.4 Hz, H-8b), 6.73 (1H, d, brs, H-2b), 6.72 (1H, d, *J* = 8.2 H-5b), 6.60 (2H, d, *J* = 8.5 Hz, H-2c,6c), 6.56 (1H, d, *J* = 1.8, H-14b), 6.52 (2H, d, *J* = 8.5 Hz, H-3c,5c), 6.30 (1H, d, *J* = 1.8, H-12b), 6.27 (1H, d, *J* = 1.8, H-12c), 6.08 (2H, d, *J* = 1.8 Hz, H-10d,14d), 6.06 (1H, brs, H-12d), 6.05 (1H, d, *J* = 1.8, H-14c), 5.96 (1H, t, *J* = 1.8, H-12a), 5.90 (2H, d, *J* = 1.8 Hz, H-10a,14a), 5.45 (1H, d, J = 5.0 Hz, H-7c), 5.39 (1H, d, *J* = 5.6 Hz, H-7d), 5.33 (1H, d, *J* = 5.6 Hz, H-7a), 4.41 (1H, d, *J* = 5.6 Hz, H-8a), 4.41 (1H, d, *J* = 5.6 Hz, H-8d), 4.21 (1H, d, *J* = 5.0 Hz, H-8c)	[74]

HPLC–DAD/FLD: high-pressure liquid chromatography-photodiode array detector/fluorescence detector; LC–MS: liquid chromatography-mass spectrometry; HPLC–DAD–FLD–MS/MS: high-pressure liquid chromatography–photodiode array–fluorescence detector–mass spectrometry/mass spectrometry; HPLC–DAD–ESI–MS/MS: high-pressure liquid chromatography-photodiode array detector–electrospray ionization–mass spectrometry/mass spectrometry; UHPLC–DAD/ESI–Q-TOF: ultra high-pressure liquid chromatography–photodiode array detector–electrospray ionization-triple quadrupole; HPLC–NMR: high-pressure liquid chromatography–nuclear magnetic resonance.

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
