# Peer review of "Grapevine Cane Extracts: Raw Plant Material, Extraction Methods, Quantification, and Applications"

_biomolecules, 2020, doi:10.3390/biom10081195_

Round 1

Reviewer 1 Report

First, I would like to make a few remarks. It is a very difficult task to write any review so that it includes all the important references to the topic, has been properly focused and only the unimportant things have been omitted. It is an even more difficult task to write a review in an area that is developing dynamically, which is precisely the processing of waste from viticulture. But I must say that the team of two experienced authors and one young researcher has succeeded. The review is well written, but there are some exceptions – small problems. The authors describe in detail the methods of extraction and analysis of the obtained samples, mentioning the applications that have applications in viticulture which are listed in the review abstract. Other possible applications, e.g. in food, cosmetics, medicine, are not listed, but I understand that the scope of the review is limited in some way.

First of all, I would like to comment the more fundamental problem of the manuscript which is the nomenclature. As the authors write on the line 48, there is no uniform nomenclature for vine shoots, which may be a task more for the OIV, but there is a clear tendency in the literature to use grapevine cane (s) for spring shoots, which are studying and discussing in most publications. Grapevine shoots are used only for just growing shoots - green - correctly shoots the vine (summer shoots) and grapevine canes for last year's vines (annual, mature woody shoot of the vine). Using terms shoots, which are more like summer shoots, could lead to a false interpretation of the results. My recommendation is to change the shoots to cane (s) throughout the whole review and check the literature data (Material and methods) for summer shoots data and make appropriate correction if necessary.

In Fig. 1, the authors present structures where the number of resveratrol units (molecules) is not correct, e.g.:

Amurensin B …it is not a dimer

Amurensin G … it is not a dimer

α – Viniferin … it is not a dimer

If trans (cis) can be given, the figure needs to be completed (ε-viniferin, ω-viniferin).

Lines 83-89…some important references from this field are missing, e.g.:

Richard T., et al.: Biochimica et Biophysica Acta 1830, 5068-5074 (2013)

Vion E., et al.: Molecular and Cellular Neuroscience 88, 1-6 (2018)

Lines 179-183…The paragraph should be completed with the effect of mechanical stress (Billet K., et al.: Food Chemistry 240, 1022-1027, 2018) and with the effect of storage conditions including special stress conditions and elicitors (Soural I., et al.: ACS Sustainable Chem. Eng. 7, 19584-19590, 2019).

Line 605…There must be mentioned very important note for the readers that filters from PVDF material absorb very selectively trans-ε-viniferin from the mixture (Morel-Salmi C., et al.: Chromatographia 77, 957-961, 2014).

Minor corrections:

Line 39… Wrong number, should be 24 144 400 000 litres, or 24 144 400 hl

Line 443…Table 3…t-ε-viniferin (t in Italic)

Line 596… hydroxycinnamoyltartaric acid

Line 977…Reference 110… The title of the article must be in lower case letters as the other references are

Line 983…Reference 112… The title of the article must be in lower case letters as the other references are

Line 992…Reference 116…The title of the article must be in lower case letters as the other references are

Conclusion

After making the proposed minor changes, I recommend the review for the publication.

Reviewer 2 Report

The Authors report a very deep review on the state of art of “Grapevine shoot extracts: raw plant material, extraction methods, quantification, and applications”, and possible perspectives are also stated.

I agree the publication of the review after minor revisions:

  • a general English language revision, throughout the manuscript (may be by an English mother tongue)
  • Figure 1B. The structures are too small (low quality) and the figure is completely un-readable (particularly the part B after the Table). Each structure should be at least as big as the trans- and cis-stilbene reported in the part A.
  • Page 10 is a white page (??)

Reviewer 3 Report

The present review highlights an interesting topic to be included in the “Biomolecules from Plant Residues” Special Issue. The extraction of added-value compounds such as stilbenes from grapevine shoots, the major by-product of vineyards resulting from the collection process, represents a promising strategy to implement a more sustainable and circular bioeconomy in the vineyard. The authors explore the state of the art regarding the stilbene composition of grapevine shoots, the extraction methods available, the analytical techniques to determine the exact extract composition, as well as the potential applications associated to stilbenes beneficial properties. In general, the manuscript is well-written, easy to follow, and provides relevant and up to date information on the topic. I believe that this review is a nice contribution to this field of research, which may be considered for publication after major revision. Therefore, I suggest that some changes may be carried out in order to improve the manuscript:

  1. Introduction:

I am not sure that it is relevant to explore so much information about pomace and lees. If these are used for further applications that could be an example for the potential of using winery by-products this should be referred. If not, I suggest reducing the information on this and focus on grapevine shoots.

The indication of the anti-inflammatory properties of stilbenes is somehow repetitive (line 86 and line 90 again). Moreover, the introduction is mainly focused in the possible application of stilbenes in the health sector. However, when we get to the application section (5) these are mainly focused in the winery field. I suggest changing these potential applications on health either to the applications section (5) or to the future perspectives section (6). A brief reference of all these applications may be only referred on the introduction along with the description of stilbenes interesting properties. Regarding stilbenes properties/potential applications, the following work could also be included: Sánchez-Gómez, R., et al. "A potential use of vine-shoot wastes: The antioxidant, antifeedant and phytotoxic activities of their aqueous extracts." Industrial crops and products 97 (2017): 120-127.

  1. Raw Plant Material: Grapevine shoot:

Line 120: the most resistant genus to what? Please clarify this information.

Table 1: This is a great effort to summarize the information of grapevine shoots stilbenes composition! It would be interesting to try to represent the different stilbenes in different columns with the respective quantification for each species.

Figure 1: A better quality is needed for publication. The representations on the part B of the figure are probably too small. Try to re-organize the figure for a bigger size. Also, if possible separate by dimers, trimers, and tetramers.  

Line 135: I suggest rephrasing to: This variability may occur due to a wide range of factors, including variety, cultivation zone, grapevine management, or extraction method.

Lines 142-155 and 156-162: it is somehow repetitive. I would recommend gathering all the related information in a single paragraph. And again, resistant to what?

Figure 2: The figure lacks quality and references should be included in the legend. Is this the same information present in Table 2? If so, I would not include this figure in the manuscript.

A conclusion paragraph could be drawn regarding the richness of different grapevine species in stilbenes (independently of the extraction/quantification method used) and the use of methods that could be applied to increase stilbene contents in grapevine shoots and add further value to this by-product. Defining which are the stilbenes with greater interest would be interesting and in which species they could be find in higher concentrations.

  1. Extraction Methods:

On this section there is great detail describing the protocols, which are also explained in Table 2. It becomes repetitive. I guess it would be more interesting to just briefly describe the methods and discuss the advantages and disadvantages of each method and the results obtained in the main text.

  1. Grapevine shoot extract analysis:

Although I think that this section is also too descriptive, I recognize that this time the methods are not described in the Table. However, some discussion/comparison on the methods available is missing and would greatly enrich this section.

  1. Applications:

Following the information in the introduction I was expecting to find information about some applications in the health field. I suggest including that information either in this section as a new sub-section or in the next one. Still regarding the health-promoting potential of stilbenes, the following studies may be explored:

  • Calvo‐Castro, Laura A., et al. "The oral bioavailability of trans‐resveratrol from a grapevine‐shoot extract in healthy humans is significantly increased by micellar solubilization." Molecular nutrition & food research 62.9 (2018): 1701057.
  • Fuchs, Christine, et al. "Polyphenolic composition of extracts from winery by-products and effects on cellular cytotoxicity and mitochondrial functions in HepG2 cells." Journal of Functional Foods 70 (2020): 103988.
  • Baechler, Simone A., et al. "Topoisomerase II-targeting properties of a grapevine-shoot extract and resveratrol oligomers." Journal of agricultural and food chemistry 62.3 (2014): 780-788.
  • Biais, Benoit, et al. "Antioxidant and cytoprotective activities of grapevine stilbenes." Journal of agricultural and food chemistry 65.24 (2017): 4952-4960.
  • Chaher, Nassima, et al. "Bioactive stilbenes from Vitis vinifera grapevine shoots extracts." Journal of the Science of Food and Agriculture 94.5 (2014): 951-954.

Line 484-486: Maybe it is important to also refer to the following work - Raposo, Rafaela, et al. "Grapevine-shoot stilbene extract as a preservative in red wine." Food Chemistry 197 (2016): 1102-1111.

Line 505: There is a recent publication that may be included in this sub-section: El Khawand, Toni, et al. "A dimeric stilbene extract produced by oxidative coupling of resveratrol active against Plasmopara viticola and Botrytis cinerea for vine treatments." OENO One 54.1 (2020): 157-164.

Line 513: significant antifungal activity in vitro or in the field?

Line 571-579: I would not use so much detail to describe the extract but rather focus on its potential.

Line 636: Remove color.

References: Check the references carefully. References number 6 and 15 are repeated e.g.

Round 2

Reviewer 1 Report

In the corrected version of the manuscript, another, fundamental error appeared, which concerns to the labelling of the spatial structure of organic compounds according to the Nomenclature of Organic Chemistry IUPAC, and which must be corrected in Table 1 and Fig. 1. The spatial structure of an organic compound is systematically characterized by one or more affixes attached to the name, which does not itself prescribe a stereochemical configuration.

Stereoisomers which differ only in the position of the atoms relative to a given plane, these atoms being or as if they were part of a rigid structure are referred to as cis or trans stereo descriptors, or in some cases (E) or (Z). These descriptors are written in italics and, in the case of (E) or (Z), not only in italics but also in capital letters in parentheses, followed by a hyphen. Descriptors (E) or (Z) are preferred when the atoms or groups are on one or more double bonds. In case of the manuscript, the descriptors cis or trans are recommended for stilbene oligomers, possibly in the abbreviation c or t.

In line 38 is still the wrong number. There should be 24 144 400 000 liters or 241 444 000 hl according OIV, the reference is not in English compare to the previous version of the manuscript.

The authors confused it together in a corrected manuscript and mix, for example, t and the lowercase letter z. I recommend strictly adhering to the descriptors cis and trans, or abbreviations not only in the Table 1 and Fig. 1, but wherever this error has occurred.

Author Response

I attach the reply to the reviewer 1.

Reviewer 2 Report

I agree for the pubblication

Author Response

The reviewer 2 has not additional quaestions.

Reviewer 3 Report

I believe that the authors did a great work to improve the manuscript, specially regarding sections 3 and 4 and the inclusion of a health-related applications subsection. However, I still have a few minor suggestions to further improve the manuscript for publication:

  1. Introduction:

For me, the connection with pomace and lees is still somehow difficult to follow. I would suggest changing the order a little:

“It is possible to distinguish two main categories of winery waste: that generated during the collection or that resulting from the winemaking process. During winemaking—in the first stage for white wines and after alcoholic fermentation for reds—must/wine is crushed in a pneumatic press producing a solid residue known as pomace. The amount of pomace generated depends on the grape variety, the cultivation conditions, and the pressing conditions used, but many researchers have concluded that pomace represents around 20%‐ 30% of grape weight [10,11]. Another kind of winemaking waste are lees. They consist of bacterial biomass, undissolved carbohydrates of cellulosic nature, lignin, proteins, phenolic compounds, tartrates acid salts and fruit skins, grains, and seed in suspension [12] produced in the tanks during the alcoholic fermentation process. Wine lees are usually at a concentration of 5% v/v [15] and they are distilled to recover ethanol or elaborate distilled beverages [16].

In the collection process, the major by‐product of vineyards is grapevine canes (also called stems, shoots or stalks), with an average production of around 2 and 5 tons per hectare and year [8]. Grapevine canes are rich in lignin, cellulose, nitrogen and potassium, the reason why they are highly composted in the field or burned [9]. However, these also present high contents of interesting compounds such as polyphenols, proteins, and stilbenes [13]. (…)”

I suggest joining information from line 99-106:

“Furthermore, stilbenes present significant anti‐inflammatory activity in the brain, which represents crucial progress in the treatment of neurodegenerative diseases such as Alzheimer’s [26–28]. Stilbenes’ potential anti‐inflammatory activity is based on the inhibition of enzymes that activate cytokines [29]. These results have important cardioprotective applications, which have been suggested in the so‐called ʺFrench paradoxʺ, which explains the low incidence of coronary heart disease among French people consuming a diet rich in saturated fats but with a high consumption of wine (a source of stilbenes) [30,31].”

  1. Raw Plant Material: Grapevine Cane

Careful with the confusion between genus and species! Moreover, I think that reordering the text starting in line 172 as follows allows to avoid repeating ideas, as previously suggested:

“The 1000 stilbenes described in the plant kingdom [45–48] have been isolated from diverse plant families such as pine (Pinaceae), cypres (Cyperaceae), peanut (Fabaceae), sorghum (Poaceae) or grape (Vitaceae) [15,49,50]. The Vitaceae family is formed of 900 species with 17 genera [51,52], and stilbenes have mainly been described in five of those genera: Ampelopsis, Cissus, Cyphostemma, Parthenocissus and Vitis. The Vitis genus is one of the most studied genera due to its economic impact. Research has identified stilbenes in different Vitis species and diverse plant parts. A summary of the identified and quantified stilbenes from grapevine canes can be found in Table 1 [57].

The composition and concentration of stilbenes are highly related to the Vitis species and cultivar aspects (growing conditions, ultraviolet irradiation, mechanical injury, chemical presence, etc.). For example, in the same climate conditions, the most resistant genus to diseases show the highest concentration of stilbenes since they are involved in the resistance response of plants to pathogen infection [53,54]. Moreover, growing and climate adversities usually result in an increase in the stilbene concentration because they are phytoalexins, and are thus produced as a response to stressful conditions [55,56].

Vitis vinifera is one of the best‐known species because it has been used in a variety of applications for years. Many stilbenes have been characterized in it, and a huge range of concentrations has been described. This variability may be due to a wide range of factors, including variety, cultivation zone, grapevine management or extraction method [73–75]. Moreover, different stilbenes were found at different concentrations in winter‐buds, shoot internodes, surface roots of the rootstock, shoot tips with young leaves and a tendril at the beginning of the leaf expansion, inflorescence at full bloom, clusters at veraison, cluster stems, grapevine canes, and berry skins and seeds from Vitis Vinifera [13]. The chemical structure of these stilbenes can be found in Figure 1.

Altogether, comparing the stilbene concentration according to the different Vitis is complicated as the grapevine canes were not extracted and analyzed under the same conditions in all the cases. Moreover, some stilbenes have not been quantified and the same stilbenes have not been studied in all the species. However, the following conclusions can be drawn from Table 1: V. amurensis presented a high concentration of total stilbenes, mainly trans‐piceatannol, trans‐resveratrol, and ampelopsin E; V. rupestris also showed a high stilbene content, with a high ω‐viniferin concentration; V. riparia stood out for its high ε‐viniferin content. In fact, V. amurensis, V. ruperstris, and V. riparia have exhibited the highest stilbene concentrations, related to the fact that they are the most resistant genera to fungal diseases [53].

Regarding grapevine canes, numerous varieties of Vitis vinifera have been examined and many extraction methodologies have been tested. For this reason, a range of concentrations have been included in Table 1, but specific data about stilbene content according to the extraction methodology and variety can be seen in Table 2. Despite the variability found, some varieties may be suggested as the highest stilbene producers, such as Pinot noir and Gewurztraminer as suggested by Guerrero et. al [72], in agreement with other authors [13,71,75–77] (Figure 2).

(…)”

Table 1: My suggestion was something similar to the Table present in the last page of this document. However, both of the formats presented by the authors are acceptable for publication. Just a note: in the text reference to Table 1 is associated only to reference 57 (line 186) but then several references are presented in the Table. Please clarify.

I suggest joining paragraph 271-274 to the previous one (266-270) as these are on the same topic.

Stilbene

Concentration (mg Kg-1 DW)

V. amurensis

V. arizonica

V. berlandieri

V. betulifolia

V. X champinii

V. cinerea

V. coignetiae

V. davidii

V. X doaniana

V. flexuosa

V. heyneana

V. labrusca

V. mustangensis

V. pentagona

V. riparia

V. rupestris

V. thunbergii

V. vinifera

t-/z-Astringin

367.0

±17.0

t-Piceatannol

1962.2

±122.4

1100.8

±21.5

894.4

±6.8

194.1

±13.6

599.7

±71.8

1195.1

±73.6

trace

1151.1

±196.5

377.0

±34.3

199.6

±34.9

615.4

±18.1

1182.7

±26.7

1710.0

- 4.0

t-/z-Piceid

*

64.2

±2.5

101.0

±1.5

133.8

±8.6

201.0

±25.0

291.7

±11.6

257.0

±9.6

284.0

- 36.0

Pterostilbene

547.0

±124.0

t-/z-Resveratrol

5432.9

±208.6

2412.4

±7.5

1951.2

±86.0

191.0

±2.4

2534.9

±15.0

3165.7

±131.3

25.0

±4.0

1049.0

±138.0

3572.2

±76.7

839.0

±31.0

1666.1

±9.4

3966.5

±52.5

*

66200.0

- 11.0

Author Response

I attach the reply to the reviewer 3.
